# Emerging Roles for Browning of White Adipose Tissue in Prostate Cancer Malignant Behaviour

**DOI:** 10.3390/ijms22115560

**Published:** 2021-05-24

**Authors:** Alejandro Álvarez-Artime, Belén García-Soler, Rosa María Sainz, Juan Carlos Mayo

**Affiliations:** 1Departamento de Morfología y Biología Celular, Redox Biology Unit, University of Oviedo, Facultad de Medicina, Julián Clavería 6, 33006 Oviedo, Spain; alejandroalvarezartime@gmail.com (A.Á.-A.); belengaso.ies@gmail.com (B.G.-S.); sainzrosa@uniovi.es (R.M.S.); 2Instituto Universitario de Oncología del Principado de Asturias (IUOPA), Santiago Gascón Building, Fernando Bongera s/n, 33006 Oviedo, Spain; 3Instituto de Investigación Sanitaria del Principado de Asturias (ISPA), Avda. Hospital Universitario s/n, 33011 Oviedo, Spain

**Keywords:** adipose tissue, browning, prostate cancer

## Abstract

In addition to its well-known role as an energy repository, adipose tissue is one of the largest endocrine organs in the organism due to its ability to synthesize and release different bioactive molecules. Two main types of adipose tissue have been described, namely white adipose tissue (WAT) with a classical energy storage function, and brown adipose tissue (BAT) with thermogenic activity. The prostate, an exocrine gland present in the reproductive system of most mammals, is surrounded by periprostatic adipose tissue (PPAT) that contributes to maintaining glandular homeostasis in conjunction with other cell types of the microenvironment. In pathological conditions such as the development and progression of prostate cancer, adipose tissue plays a key role through paracrine and endocrine signaling. In this context, the role of WAT has been thoroughly studied. However, the influence of BAT on prostate tumor development and progression is unclear and has received much less attention. This review tries to bring an update on the role of different factors released by WAT which may participate in the initiation, progression and metastasis, as well as to compile the available information on BAT to discuss and open a new field of knowledge about the possible protective role of BAT in prostate cancer.

## 1. White Adipose Tissue (WAT)

In adult mammals, WAT is widely distributed throughout the body to provide one of its main functions, i.e., offering mechanical protection against impacts, thus preventing organ harm. Additionally, it also serves as a long-term energy store by accumulating triglycerides in the form of lipid droplets inside the adipocytes [1], as the complete oxidation of fatty acids provides more than twice the energy associated with carbohydrates and proteins [2,3]. Therefore, white adipocytes, namely unilocular adipocytes, may exceed 100 μm in diameter and usually display a large droplet surrounded by a phospholipid monolayer [4], which restricts the nucleus and cytoplasm to the periphery, leaving less than 2 μm between the droplet and the plasma membrane (Table 1) [5].

Two genes are responsible for many of the changes observed during WAT adipogenesis, i.e., the proliferative activated peroxisomal receptor (PPARɣ) and the α-protein binding sequence CAAT (C/EBPα) [6]. PPARɣ is essential for the differentiation process to occur and in turn, it also induces the expression of C/EBPα, which makes this gene a key regulator of WAT differentiation [7]. On the other hand, it has been shown that C/EBPα expression is necessary but not sufficient, since C/EBPα^−/−^ cells are capable of proceeding through the differentiation process. However, this differentiation is incomplete since the cells store a lower lipid content. Once the preadipocyte reaches a ‘threshold’ in the expression of these two genes, cells begin to accumulate triglycerides in small lipid droplets, which eventually merge into a single one.

## 2. Brown Adipose Tissue (BAT)

In mammals, BAT is also widely distributed but restricted to the first stages of development, tending these fat depots to disappear with age. However, this is not the case for all species, since this specialized adipose tissue prevents the body temperature drop that would occur in many heterotherms during hibernation. Classically, it has been assumed that human adults show small BAT depots limited to certain regions including the cervical, supraclavicular and axillary areas, perivascular regions and small areas close to the kidney. Nevertheless, very recently, Leitner and co-workers have accurately mapped the human BAT depots, using positron emission tomography/computed tomography, showing that previous studies have underestimated the presence of this tissue in adults [8].

BAT major function relates to the ability to produce the so-called ‘non-shivering thermogenesis’, a process that, at the molecular level, is carried out by the mitochondrial uncoupling protein 1 (UCP-1/thermogenin) [9]. As its own name suggests, this transmembrane protein uncouples the ETC by pumping protons from the intermembrane space back into the mitochondrial matrix, thereby generating heat rather than ATP [10]. This protein is activated by allosteric binding of long-chain fatty acids produced during the lipolysis of cytoplasmic lipid droplets after adrenergic stimulation [11,12]. Furthermore, it is also subjected to positive transcriptional regulation by different factors, both genetic such as the presence of a 211-base pair enhancer, and molecular such as rosiglitazone or retinoic acid [13,14].

BAT adipocytes are characterized by a central nucleus and a cytoplasm filled with numerous small lipid droplets (multilocular adipocytes) as well as many mitochondria which are important for the thermogenic process, giving the tissue its characteristic brown color. Like WAT, BAT mainly accumulates triglycerides (30–70%), but in this case, brown adipocytes may also accumulate phospholipids such as phosphatidylcholine, phosphatidylethanolamine, and, to a lesser extent, phosphatidylinositol. Many of these features differentially displayed by adipocytes are presented in Table 1 [15].

**Table 1 ijms-22-05560-t001:** Main differences among the three types of adipose tissue. Modified from Cedikova. et al. [16].

Characteristic	White	Brown	Beige
Morphology Shape	Spherical	Polygonal	Spherical
Cell Size	Variable, large	Small	Variable
Lipid Droplet	Single large LD	Multiple small	Multiple variable
Mitochondria	+	+++	++ (upon stimulation)
Development	From Myf 5^−^ precursors	From Myf 5^+^ precursors	From Myf 5^−^ precursors
Location	Subcutaneous Visceral	Cervical Supraclavicular Axilar	Inguinal Perivascular Other locations?
Function	Energy storage	Heat production	Adaptative thermogenesis
Uncoupling Protein	Nearly undetectable	+++	++ (upon stimulation)
Vascularization	Low	High	High (upon stimulation)
Lipid Content	Tryglicerides	Tryglicerides phopholipids	Tryglicerides phospholipids
Impact on Obesity	Positive	Negative	Negative

## 3. ‘Beige’ Adipose Tissue

In recent decades, a phenomenon known as ‘browning’ of WAT has been described, which was first reported by Young and colleagues in 1984 [17]. This process is triggered by the increased gene expression levels of different markers involved in the BAT adipogenic differentiation, including PPARɣ or PGC-1α [18]. Such increase causes the appearance of multilocular adipocytes, morphologically similar to brown adipocytes, but rather located in those anatomical sites characteristic of WAT [19]. PPARɣ, a key regulator of mitochondrial biogenesis and activator of oxidative metabolism, also induces the expression and production of UCP-1 in these adipocytes [20]. Other genetic markers of the browning process are the bone morphogenetic proteins 4 and 7 (BMP4, BMP7), which act as increasing both energy expenditure and insulin sensitivity in adipocytes [21]. Browning mainly occurs due to non-pharmacological activators such as thermal stimuli including exposure to low temperatures, physical exercise or diet [22,23]. Nevertheless, browning can be pharmacologically induced by drugs that trigger the β-adrenergic response, or by modulation of proteins involved in the adipose tissue differentiation such as PPARɣ, AMPK or Sirtuins [24]. These stimuli cause sympathetic nervous system activation, which in turn triggers β-adrenergic signaling mediated by norepinephrine (NE) [25]. Browning of male murine subcutaneous WAT after orchidectomy has also been described, though the mechanistic insight of this process is still unclear [26].

Even though beige adipocytes display some ultrastructural and biochemical features in common with brown adipocytes, such as mitochondrial content and elevated UCP-1 expression, both types of adipocytes differ in many features [27]. While brown adipocytes derive from a mesenchymal precursor expressing myogenic factor 5 (Myf5) and paired box protein 7 (Pax7) as cellular markers, beige adipocytes have a controversial origin. Some experts defend that these cells would originate from WAT transdifferentiation [28]. This alteration is supported by evidence indicating that, in nursing mothers, breast adipose tissue is capable of transdifferentiating into glandular tissue and then dedifferentiate back into mature adipose tissue after the completion of nursing [29]. On the other hand, some groups defend that beige adipocytes would arise from de novo differentiation of mesenchymal precursors residing within the adipose tissue stroma [30] (see Table 1).

Preliminary results from our group agree with the phenomenon of transdifferentiation from WAT adipocytes, since we observe an increase in the amount of beige adipose tissue between the periprostatic adipose tissue depots (PPAT), without altering the total fat content in the prostate, a finding not previously described in the literature (unpublished data). The browning process has been associated with different roles in obesity, where a beneficial effect has been reported by increasing energy expenditure in favor of thermogenesis, thus decreasing the amount of WAT [26]. On the other hand, the browning process might have a role in the development of breast tumors or participate in collateral phenomena such as cachexia [31].

## 4. Prostate Cancer: A Key Role for Adipose Tissue

Prostate cancer is the most commonly diagnosed tumor among men in western countries and the second most common in the world after skin cancer. Its incidence is determined by several factors such as age, race or lifestyle [32]. The latter is currently thought to have a key role in its development and progression. During the last years, an increase in prostate cancer incidence has been observed, positively correlated with an increase in the prevalence of other pathologies such as obesity or metabolic syndrome [33]. It has been widely reported that a high-fat diet is related to an increase in the number of metastases originating from a primary prostate tumor [34]. As a consequence, a variety of mechanisms have been elucidated supporting the relationship between these pathologies, including elevated systemic and local inflammation levels derived from increased infiltration of immune cells in fat depots, hyperinsulinemia as a consequence of the increase in adipose tissue and its activity, or alterations in the availability of lipids and adipokines [35]. All these mechanisms have been proposed, to a greater or lesser degree, as keys in the carcinogenesis process of different tumors and have been reviewed in detail elsewhere [36].

Furthermore, WAT is one of the most abundant components of the prostate microenvironment both in healthy and tumorous settings and its activity is closely related to the gland. Given its endocrine role, WAT modulates the activity of both, glandular cells and the microenvironment itself causing alterations that favor the processes of migration, invasion and aggressiveness of prostate tumors [37]. However, this interaction is not unidirectional, since several studies have revealed the capacity of the prostate tumor to induce genetic and phenotypic changes in adjacent adipocytes, leading to their transformation into the so-called ‘cancer-associated adipocytes’ [38].

## 5. Adipose Tissue-Tumor Crosstalk. A Role for BAT in Tumorigenesis?

In normoweight subjects with BMI between 18.5–25, the microenvironment of adipose tissue is rich in anti-inflammatory cytokines produced by the immune cells found in the extracellular matrix and also secreted by the adipocytes themselves [39]. However, when weight gain occurs, in addition to the secretion of adipokines related to insulin resistance or inflammation, the adipocytes become hypertrophied and, in some cases, their own cell death causes the release of cellular components as well as intracellular molecules that will ultimately trigger the innate immune response activation. Consequently, the microenvironment acquires a pro-inflammatory profile contributing to the recruitment of more immune cells that feed back the process [40]. It is described that both, induction and transplantation of BAT in individuals with adipose tissue dysfunction have a beneficial role in reducing weight gain [41]. Stanford and colleagues have described how BAT transplantation is able to regulate metabolic homeostasis, thus improving glucose tolerance and insulin sensitivity [42]. Similarly, Vargovic and co-workers observed how cold-induced BAT exerts anti-inflammatory effects, which were manifested through an upregulation of anti-inflammatory cytokines, i.e., IL-4 and IL-10 families, as well as a downregulation of pro-inflammatory cytokines of the IL-1β, IL-6 or IL-17 families [42]. Furthermore, they also suggest an increase in the number of polarized macrophages (M2) in the adipose tissue microenvironment to the detriment of classical activated macrophages (M1) in WAT dysfunction associated with obesity. These findings are in line with data published by others groups, which support the anti-inflammatory profile of BAT [43].

Cancer-associated adipocytes have been shown to be ultimately capable of producing a vicious feedback loop that increases the malignancy of tumor cells. To promote this malignancy, adipocytes and cancer cells establish crosstalk mediated by paracrine signaling. This interplay is based on the release of different adipokines that modulate cell growth and survival, as is shown in Figure 1. Leptin and adiponectin are two of the most important hormones involved in this process and are also the main adipokines released by adipose tissue [44]. In normal prostate, adipose tissue is present in approximately 48% of the gland surface [45]. During tumor growth, epithelial cells leave the glandular niche and invade the microenvironment becoming exposed to huge concentrations of adipokines released by periprostatic adipose tissue that boost their proliferation [46]. Likewise, it has been shown in vitro that co-culture of human prostate tumor cells, LNCaP and PC3, with adipocytes favors the epithelial-mesenchymal transition and weaponize tumor cells with certain resistance to chemotherapy treatment such as docetaxel or cisplatin [47]. In addition, a study using ob/ob leptin deficient mice, which develop obesity due to lack of the aforementioned hormone, found that the induction of benign prostatic hyperplasia (BPH) by exogenous testosterone administration resulted in an attenuated phenotype in comparison with controls and ob/ob mice [48]. Moreover, ob/ob mice treated with testosterone showed higher levels of E-cadherin and lower levels of vimentin, suggesting that leptin plays a key role in the EMT [49]. In accordance with these results, other groups found that the treatment of high fat diet mice with exogenous recombinant adiponectin, the opposite of leptin, protects obese mice against BPH [50].

In human studies, Sacca and colleagues described that the PPAT secretome of different stages of prostate cancer and BHP contains proteins involved in lipid transport and adipogenesis. The levels of these proteins were higher in advanced stages of tumors and decreased in early stages, with lower levels in patients with BPH. According to these results, researchers related this secretome with some pathways involved in cell adhesion, migration and invasion [51].

These alterations make this crosstalk become a key factor, due to its influence on tumor development and growth during the initial stages of tumorigenesis, which has subsequent consequences in the development of the disease and its mortality.

### 5.1. Leptin

Leptin is a hormone released by adipose tissue, involved in the regulation of food intake and body weight through hypothalamic control [52]. Commonly known as the ‘satiety hormone’, leptin is responsible for reducing the appetite sensation when reserve levels in adipose tissue are high, thus favoring energy expenditure [53,54]. In tumors, however, leptin acts rather as a promoter of growth and cell proliferation, by ultimately leading to phosphorylation of the mitogen-activated protein kinases (MAPKs), as well as acting as a cyclin D1 promoter, which collectively contributes to the progression of the cell cycle [55,56]. In obese subjects, the levels of this hormone are higher than in non-obese individuals. Epidemiological studies have found an association between high levels of leptin and an increased risk of colon and breast cancer [57,58].

In prostate cancer, its effects have been demonstrated in cell culture models, where leptin stimulation increases cell growth in androgen-independent PC-3 and DU145, but not in hormone-responsive LNCaP cells [59]. Regarding the relationship between circulating levels of leptin and prostate cancer, there are some conflicting results. While some groups describe that increases in serum leptin levels show a positive correlation with prostate cancer risk, other groups report no differences between them [60,61]. It is noteworthy to mention that such association between high levels of circulating leptin and prostate cancer is more striking in advanced prostate tumors, thus its use as a potential tumor progression marker should not be ruled out.

### 5.2. Adiponectin

Adiponectin is a peptide hormone mainly synthesized by adipose tissue whose main role resides in regulating glucose metabolism, by promoting insulin sensitivity and glucose uptake and stimulating fatty acids oxidation [62]. Adiponectin interacts with cells through its membrane receptors, namely ADIPOR1 and ADIPOR2 [63]. Upon interaction with their membrane receptors, it triggers cellular signaling that involves antitumor effects including the induction of caspase-mediated apoptosis, anti-inflammatory activity mediated by nuclear factor kappa-light-chain-enhancer of activated B cells (NFκB), or anti-angiogenic activity by the inhibition of vascular endothelial growth factor A (VEGF-A) [64,65]. Furthermore, adiponectin is found at very low levels in obese individuals and shows a negative correlation between its circulating levels and the occurrence of some tumors, making this adipokine a prognostic risk factor to be considered [53]. Both adiponectin receptors are overexpressed in invasive and non-invasive breast tumors [66]. On the contrary, in the prostate, the presence of the ADIPOR1 receptor is diminished in tumor tissue, compared to non-pathologic prostatic tissues from healthy individuals [67]. Similar results have been reported in colon cancer, where adiponectin levels have been negatively correlated with the development and progression of neoplastic lesions [68]. In breast cancer, another endocrine-related tumor, adipose tissue plays an important role. Breast fat depots are the major endocrine system of the breast and release a large variety of growth factors and enzymes that contribute to the normal development, differentiation and homeostasis of the gland [69]. It has been described an abnormal development of the breast acinus in the absence of leptin that it is recovered when leptin was re-administrated [70]. Furthermore, excessive growth of the mammary gland was observed in mice when adiponectin was overexpressed [71].

In addition, adipose tissue differentiating genes such as PPARɣ and C/EBP are considered tumor suppressor genes in breast cancer. Evidence has shown that downregulation of PPARɣ correlates with a protumorigenic ambient. C/EBP was related to estrogen negative and metastatic breast cancer [72,73].

According to the secretory activity of adipose tissue, breast adipose tissue hormones also play an important role. Leptin activates the MAPK and AKT pathways increasing the proliferation, angiogenesis and decreasing apoptosis in tumor cells [74]. Moreover, leptin activates the estrogen receptor and the epidermal growth factor receptor 2 (HER2) which results in an increase in cell growth and resistance to therapies [75]. On the contrary, adiponectin acts as an antagonist, showing antiproliferative and proapoptotic effects by inactivation of AKT and activation of STAT3 pathways [76].

On the other hand, given the high functionality of BAT in the control of energy homeostasis, its possible influence on tumor processes has been studied. Experiments carried out mostly in breast cancer, relate an increase in the expression of anti-tumor genes such as phosphatidylinositol 3,4,5-trisphosphate 3-phosphatase (PTEN) with the presence of BAT in WAT depots in vivo [77]. PTEN is a gene capable of regulating cell growth and proliferation by controlling the cell cycle [78]. However, the presence of brown adipose tissue might negatively affect the survival of cancer patients due to its role in the phenomena of cancer-associated cachexia, since it contributes to a rapid loss of fat and the dysfunction of the adipose tissue itself due to its high metabolic activity [79]. Collectively, the opposing data reported to date point that the role of this type of adipose tissue within the tumor process remains unclear and many of its routes of action are unknown, therefore it requires further study and further investigation into its possible function. Nevertheless, to date, several factors derived from adipose tissue have been reported to have a role in cancer progression, related to extracellular matrix remodeling and/or metastasis.

## 6. Adipocytes Modulate Extracellular Matrix Reorganization

The extracellular matrix (ECM) is a complex structure made up of different proteins, proteoglycans and polysaccharides that serves as a scaffold for cells. This structure supports cells in processes such as cell adhesion, tissue repair and regeneration, or survival [80].

In obesity, ECM remodeling is crucial, since it needs to undergo a reorganization to allow accommodation of the hypertrophic adipocytes, which makes it highly plastic based on fluctuations in the lipid content of adipocytes [81]. Likewise, during tumorigenesis, ECM remodeling is a key factor involved in the progression, since the undergone alterations allow invasion and cell migration [82].

Endocrine activity of adipocytes includes several factors implicated in ECM formation and remodeling. Therefore, during tumorigenesis, tumor cells switch on the synthesis and release of molecules such as collagenases or metalloproteinases, favoring a better environment for cell migration out of the primary tumor [45,83]. In addition, as mentioned above, an exaggerated increase in adipose tissue leads to the recruitment of white blood cells that will ultimately trigger local inflammation due to the release of proinflammatory cytokines such as IL-1, IL-6 or TNF-α both by adipocytes and immune cells [84].

### 6.1. Interleukin-1 (IL-1)

IL-1 is a regulatory cytokine involved in different processes, including host defense response to inflammation and fever. Its pivotal role in fever after the activation of peripheral immune cells has been widely reported [85]. Additional cellular functions including its regulatory functions on cell growth and survival through the mediation of NFκB, upon binding to its receptor, have also received attention [86].

This cytokine was found elevated in the plasma of obese subjects [87,88]. Furthermore, it is described that circulating levels of IL-1 would be a risk factor in various tumor types such as lung and colorectal cancer [89,90]. In the tumor microenvironment, not only is IL-1 released by the activated macrophages present in the area but it is also produced by adipose tissue [91]. Squamous carcinoma cells either, produce IL-1 or stimulate cells of the tumor microenvironment to produce it in order to be later used by tumor cells [92]. IL-1 effects on growth and progression have been described in most of the known tumors [93]. In the case of prostate cancer, several laboratories have reported in vitro effects on the modulation of neuroendocrine differentiation processes in the prostate, a phenomenon that increases the aggressiveness, chemo/radioresistance and worsens tumor prognosis. Furthermore, it can drive the migration of tumor cells out of the glandular niche [94,95,96].

### 6.2. Interleukin-6 (IL-6)

IL-6 is a cytokine produced by adipose tissue whose levels have been seen increased in obese individuals [97]. This molecule is an important modulator in the chronic inflammation associated with carcinogenesis, and its tumorigenic properties are related to both, the Janus kinase 2 and signal transducer and activator 3 pathway (JAK2/STAT3) and the phosphoinositide 3 kinase and serine/threonine kinase pathway (PI3K/Akt). Both kinases are essential regulators of the antiapoptotic and proliferative mechanisms in tumor cells. Consequences of IL-6 activation have been observed in a variety of tumors, such as breast cancer and colorectal cancer [67]. In prostate cancer, on the other hand, circulating levels of this cytokine are found elevated in those patients with untreated metastases or hormone-refractory tumors, when compared to individuals with a localized tumor [98]. In addition, several groups have described the ability of androgen-independent, PC3 cells and DU145 to produce this cytokine indicating a paracrine or autocrine role of the molecule itself [99].

### 6.3. Tumor Necrosis Factor Alpha (TNF-α)

TNF-α/TNFA is synthesized mainly by the monocyte/macrophage system, but it is not restricted to it [100]. This pleiotropic cytokine is one of the proinflammatory adipokines produced and secreted by adipose tissue and plays an important role in pathogenesis and disease by interacting with its membrane receptors [101]. Overweighted individuals exhibit higher TNF-α levels than subjects with normal BMI [102]. In addition, this increase in TNF-α is related to oncogenic phenomena in several tumors such as colorectal tumors [89].

In tumors, it is thought that the activity of TNF-α may be related to the tumor microenvironment at the local level of the adipose tissue, where it acts as a signaling molecule, either at the paracrine or autocrine level, regulating the different processes of tumor development and progression. In addition, this cytokine is related to the regulation of other adipokines involved in tumorigenic processes such as Interleukin 6 (IL-6) [103]. In prostate tumors, a dual function has been described for this molecule [104]. On the one hand, there is evidence that it stimulates angiogenesis and is involved in processes such as the acquisition of androgen independence, related to tumor progression and a worse prognosis [105]. However, other research groups have described different antitumor activities such as the inhibition of neovascularization or the ability to induce apoptosis in prostate tumor cells [106].

BAT has acquired in recent years the qualification as an endocrine organ, similarly to that of WAT. This statement relates to the ability of this tissue to regulate energy homeostasis in a UCP-1 independent manner. Therefore, molecules similar to those adipokines secreted by WAT have been also identified and in this case, they are termed brown adipokines or “batokines” [107]. Even though some of these batokines are common with those produced by WAT, there are, however, some specific cytokines exclusively produced by this tissue [108]. Recently, the potential protective effect of these batokines in different pathologies, including metabolic syndrome or obesity has been proposed, making them gain great importance [109]. The classical batokines originally produced by this tissue are secreted molecules that act primarily in a paracrine or autocrine manner such as vascular endothelial growth factor (VEGF), bone morphogenetic proteins (BMPs) or UCP-1 [110,111,112,113].

### 6.4. Vascular Endothelial Growth Factor (VEGF)

VEGF is widely distributed throughout the body and plays an important role in different tissues [114]. BAT is a known source of VEGF, which in turn performs important functions primarily at the autocrine and paracrine levels. It has been observed that BAT-induced increases in VEGF secretion do not significantly contribute to the blood concentration of the factor, so it acts mainly at the local level [115,116]. In adipose tissue, VEGF signaling has been associated with browning of WAT after exposure to low temperatures, as well as with decreases in the metabolic dysfunction caused by diet-induced obesity (DIO) [117]. In tumors, VEGF plays an important role in the progression and metastasis due to the de novo blood vessel formation, thus favoring growth and dissemination [118,119].

Bone morphogenic proteins (BMPs) are extracellular signaling proteins with an important presence in bone and cartilage, where they regulate the neoformation and regeneration of these supporting tissues [120]. In adipose tissue, these proteins intervene in the process of cell differentiation from the mesenchymal precursor to the mature adipocyte [121]. The main BMP produced by BAT, i.e., BMP7, stimulates the production of UCP-1 and mitochondrial biogenesis and in turn triggers subcutaneous WAT browning. However, this is not the principal actor among BMPs in adipogenesis, as BMP4 and particularly BMP8 show a more prominent role. At a systemic level, the increase in the expression and production of these proteins is related to an increase in basal energy expenditure and therefore has important beneficial effects in the protection against DIO [113].

In prostatic tumors, beneficial effects from increasing extracellular BMP7 levels against the migration and invasion of tumor cells have been reported, since it is not a protein produced in large quantities by the epithelial cells of the prostate [122]. It has been observed that in prostate tumors the levels of this protein are reduced in relation to healthy glandular tissue [123]. Several groups have described an overexpression of this protein response in relation to androgen deprivation [124]. Likewise, it has been observed that treatment with this batokine causes an upregulation of epithelial markers such as E-cadherin in contrast to a down-regulation of mesenchymal markers such as fibronectin or vimentin in prostate tumors [125].

### 6.5. Mitochondrial Uncoupling Protein 1(UCP-1)

Even though it is not a secreted protein, UCP-1 may have important effects in the regulation of the tissue microenvironment due to its indirect role in redox homeostasis [126]. Adrenergic regulation after BAT stimulation with NE modifies the secretome considerably. Under normal conditions, in the absence of adrenergic stimulation, the batokines released correspond to those proteins involved in the remodeling of the matrix, such as lysozyme C-2, collagenase alfa 1 or cathepsin. However, NE induces the production of batokines, mainly related to protein folding response, involved in various routes such as adipogenesis, lipolysis, autophagy or redox control [127]. Regarding the latter, an increase in antioxidant enzymes such as thioredoxin 1, heme oxygenase, or catalase is concomitant with an increase in PPARɣ levels [128]. The role of BAT in the protection against oxidative damage is not new. Several works have highlighted the evolutionarily conserved role of UCP-1 in reducing the production of superoxide anion by mitochondria, where it decreases the membrane potential between the inner membrane and intermembrane space, thus making it difficult for electrons to pass through the transport chain to reach the molecular oxygen [129,130]. In addition, as already mentioned above, increases in the expression levels of UCP-1 are associated with an elevated expression of PPARɣ [131].

ECM alterations also play a key role in the ‘epithelial-to-mesenchymal’ transition (EMT) process that tumor cells undergo during the early stages of invasion and migration [132]. In cells this process occurs normally under physiological conditions, especially during embryonic development, favoring the creation of different tissues and organs [133]. This phenotypic change in cells endows them with special features in which they acquire greater plasticity, proliferative capacity and migration capacity, which makes it a sign of malignancy and worse prognosis in tumor cells [134]. This process is characterized by the alteration of the expression patterns and the production of proteins involved in cell-cell and cell-matrix adhesion, such as E-cadherin or integrins, whose levels decrease in favor of mesenchymal markers including N -cadherin or vimentin [135]. For this process to occur, the cells respond to different stimuli, either intracellularly as gene expression such as WNT or Headhog or due to extracellular factors such as EGF, HGF, TGF-β, FGF or IGF present in the extracellular matrix and which are released by fat cells [136]. Table 2 shows a detailed list of the most important batokines and their reported assayed levels in solid tumors.

## 7. Adipose Tissue and the Pre-Metastatic Niche

Once the role of adipose tissue in feeding and fueling tumor cells as well as in the invasion and migration of tumor cells outside the gland has been reviewed, another important event to focus on is the implication of adipose tissue in the promotion of the metastatic process [220]. The metastatic process is one of the factors related to the increase in morbidity and mortality in cancer patients [221]. In this process, bidirectional communication between tumor cells and the microenvironment is crucial [222]. An important and necessary factor for metastasis to take place is the adequacy of the microenvironment of the target organ or tissue to the establishment of tumor cells that have escaped from the primary tumor and are in the bloodstream as circulating cancer cells. This process is known as the establishment of the pre-metastatic niche and is based on the adaptation of a permissive environment in terms of nutrients, extracellular matrix and immunity that allows the cells to stay and proliferate [223]. The high secretory capacity of adipokines by adipose tissue, and its role as one of the largest endocrine organs, makes it one of the potential tissues involved in metastasis. In addition, the increase in the levels of different cytokines in blood during obesity reveals that adipose tissue is capable of influencing other distant tissues and modulating their microenvironment in order to admit metastatic tumor cells [224].

Furthermore, another important factor in the establishment of the pre-metastatic niche is the ability of adipose cells to release extracellular vesicles (EVs), which help to promote the establishment of the premetastatic niche and subsequent colonization [225]. The EVs can be classified into two subtypes according to their size and origin: microvesicles and exosomes. Microvesicles (100–1000 nm) are a heterogeneous population of vesicles that come directly from the evagination of the plasma membrane, while exosomes (30–100 nm) derive from luminal membranes of multivesicular bodies, which through fusion events with these smaller vesicles are released from the plasma membrane [226]. EVs are involved in the horizontal transfer of material between cells modulating the phenotype and functional characteristics of cells as well as their number and content depending on the cellular and physiological activity in which the exosomes are involved. However, given their great stability in fluids such as blood, they have become emerging biomarkers with great potential for detecting pathologies [227].

Injecting exosomes of 3T3-L1 preadipocytes into breast fat depots, together with MCF10-DCIS tumor cells has been reported to promote the growth of the primary tumor [35]. Furthermore, in vitro treatment of MCF-7 cells with exosomes derived from adipose tissue mesenchymal cells favors the migration of tumor cells in a dose-dependent manner [228]. It has recently been described that exosomes secreted by adipose tissue, under physiological conditions, exhibit a specific organotropism in distant organs with mechanisms of action similar to adipokines. The surface markers that give this organotropism are not known, although it is speculated that they may be the integrins that are present on their surface [229].

The cargo of these exosomes largely varies and may contain different molecules depending on the types of fat. By using a proteomic approach, a recent report has described the presence of 262 molecules commonly secreted by WAT and BAT. Among these, a subset of 101 molecules is specifically released by BAT. This differential secretion allows discrimination of two secretory populations [230]. In accordance with these results, another group has confirmed the possibility to differentiate two populations of EVs coming from WAT and BAT [231]. WAT EVs contain some molecules related to ECM reorganization such as TGF-β, tenascin or different MMPs, together with proinflammatory cytokines that are also involved in the ECM reorganization and accommodation in the target organ [232]. Exosomes from adipose tissue loaded with specific miRNAs have been reported to regulate the expression of pathways such as TGF-β and Wnt/β-catenin in obesity models, inducing liver fibrosis [233]. Furthermore, the release of exosomes loaded with proinflammatory cytokines modulates the immune response of the target tissue microenvironment establishing an appropriate pre-metastatic niche for the nesting of circulating tumor cells [234]. On the other hand, BAT EVs’ cargo is not clearly known. Some reports propose that BAT-derived EVs have beneficial effects by causing a negative regulation of the immune innate response mediated by different molecules, including complement factor H and various anti-inflammatory interleukins such as IL-10, as well as macrophage differentiation mediated by adiponectin [230,232].

## 8. Conclusions and Future Perspectives

Obesity is clearly a widespread problem in developed countries. In addition, it is involved in other pathologies such as diabetes, metabolic syndrome or cancer. Prostate cancer is one of the most frequent tumors in men in the western world. Despite advances in the field, primary tumors eventually progress to an aggressive and metastatic phenotype but treating these tumors remains a challenge. The prevalence of obesity in the developed world and the mechanisms that associate suffering from this pathology with tumor processes are important to study. However, patients with high-grade tumors are often lipodystrophic. Furthermore, in advanced stages, tumors are associated with a loss of function and WAT quantity, as well as increased body wasting (cachexia), which difficult and compromises survival. Likewise, interactions between adipose tissue and metastatic processes have been described, which makes it necessary to expand our studies on the potential role of this tissue in the processes of invasion, establishment of premetastatic niche and metastasis. The transformation of WAT to Beige/BAT has been proposed to play a role in the process. As we have mentioned, brown adipocytes have a high energy dissipation capacity. However, a relationship has been established between BAT activation and tumor progression in relation to cachexia. Apart from these results, the research in BAT is needed to broaden the field, since its potential role in releasing anti-inflammatory cytokines that could regulate the tumor microenvironment has also been described, which in turn would improve the prognosis and palliate the progression of different tumors.

## Figures and Tables

**Figure 1 ijms-22-05560-f001:**
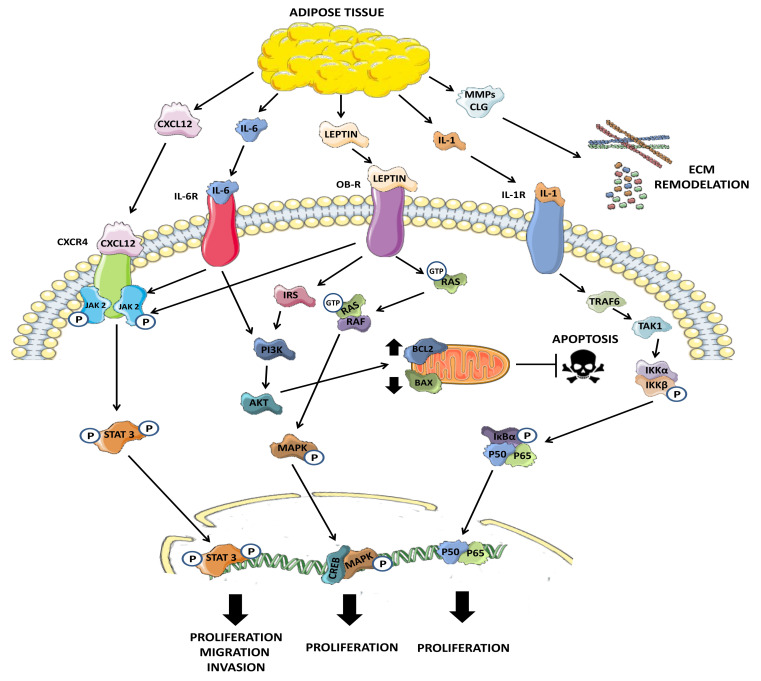
Major routes driving the influence of adipose tissue on tumor progression. Adipose tissue can release adipokines which play important roles in normally altered pathways in cancer. The stromal cell derived factor 1 (CXCL12) through direct interaction with the chemokine receptor CXC type 4 (CXCR4) present in the membrane of tumor cells activates the JAK2/STAT3 signaling cascade, that ultimately has the function of translocation from STAT3 to the nucleus where it acts as a transcription factor for genes involved in cell survival, or migration and invasion, thus promoting EMT. Leptin, another of the main adipokines in adipose tissue, acts by interaction with its membrane receptor (OB-R). This receptor is associated with the JAK2 protein, which in turn phosphorylates STAT3, triggering the activation of survival pathways. Likewise, its interaction with the serine/threonine kinase RAF protein, and the GTPase RAS induce the activation of the MAPK pathway, promoting cell growth and survival. This hormone is also important in the regulation of apoptosis since via PI3K/AKT can modulate the levels of the BCL-2 and BAX proteins, inhibiting apoptosis of tumor cells and thus favoring both proliferation and resistance to chemotherapy. In an inflammatory environment, it is common for the adipose tissue and its microenvironment immune cells to release different interleukins such as IL-6 or IL-1, which by binding to their membrane receptors (IL-6R and IL-1R) are capable to activate the survival pathways mentioned above. IL-1, can also promote proliferation and survival by activating NFκB-mediated signaling. Finally, adipose tissue can contribute indirectly to the spread of tumor cells through the remodeling of the extracellular matrix mediated by collagenases (CLG) and metalloproteinases (MMPs) which, in conjunction with EMT, favors the migration of cells out of the primary niche and their dissemination.

**Table 2 ijms-22-05560-t002:** Common reported levels of “batokines” in solid tumors.

Batokine/Gene	Physiological Activity	Cancer Type	Protein Levels ^1^	Model	References
Adiponectin/ADIPOQ	Receptor mediated: Regulates proliferation Receptor independent: Regulates insulin sensitivity and modulates immune response	Colorectal	Low (serum)	H	[137]
		Breast	Low (serum)	H	[138,139]
		Pancreas	High (serum) Low (serum)	H	[140,141]
		Liver	High (serum)	H	[142]
		Prostate	Low (serum)	H	[143,144]
		Lung	Low (advanced stages)	H	[145]
		Melanoma	Low (serum)	H	[146]
Bone morphogenetic proteins/BMPx	Receptor mediated: Regulates proliferation, differentiation and apoptosis	Colorectal	High BMP2 (metastasis) Low BMP3 (Primary)	H	[147,148]
		Breast	High BMP4, BMP7 (CL, Primary)	CL, M, H	[148,149]
		Pancreas	Low (Cells)	CL	[150]
		Liver	High (serum)	M	[151]
		Prostate	Low (Primary)	H	[152]
		Lung	BMP2 and BMP 4 high (Primary). BMP7 high (metastasis)	CL, H	[153,154]
		Melanoma	BMP7 high (Primary)	H	[155]
C-X-C Motif Chemokine Ligand 14/CXCL-14	Implicated in recruitment of immune cells and immune surveillance	Colorectal	Low (Primary)	H	[156]
		Breast	Low (Primary, CL)	CL, M, H	[157,158]
		Pancreas	High (Primary, CL)	CL, H	[159]
		Liver	Low (Primary, CL)	CL, H	[160]
		Prostate	High (Primary, CL)	CL, H	[161]
		Lung	Methylated (Primary)	H	[162]
		Melanoma	N/A	H	[163]
Retinol binding protein 4/RBP-4	Regulates insulin resistance and glucose homeostasis	Colorectal	High (serum), High (CL)	CL, H	[164,165]
		Breast	High (serum)	H	[166]
		Pancreas	High (serum, Primary)	H	[167]
		Liver	High (CL), Correlation serum levels-survival	CL, H	[168]
		Prostate	High (CL)	CL	[169]
		Lung	High (serum)	H	[170]
		Melanoma	N/A		
Fibroblast growth factor 21/FGF-21	Regulates cell proliferation, glucose homeostasis and acts as a stress sensor	Colorectal	High (serum)	H	[171,172]
		Breast	Low (serum) with hormonal therapy	H	[173]
		Pancreas	Low (Primary)	M	[174,175]
		Liver	High (Primary)	M	[175,176]
		Prostate	Low (Primary, CL)	CL, M, H	[177]
		Lung	High (Primary, CL)	H	[178]
		Melanoma	N/A		
Insulin growth factor binding protein 2/IGFBP-2	Interacts with components of ECM, controls cell growth and metabolism	Colorectal	High (serum)	H	[179,180]
		Breast	High (Serum, Primary)	CL, H	[181,182]
		Pancreas	High (pancreatic juice, serum)	H	[183]
		Liver	High (serum)	H	[180,184]
		Prostate	High (serum, CL)	CL, H	[185,186]
		Lung	High (serum), Secreted (CL)	CL, H	[187,188]
		Melanoma	High (Primary, CL)	CL, H	[189]
Insulin like growth factor 1/IGF-1	Displays activity by a receptor mediated interaction. Plays an important role in growth and aging	Colorectal	Correlates with cancer risk	H	[190]
		Breast	High (serum)	H	[182,191]
		Pancreas	High (Primary)	H	[192]
		Liver	Low (serum)	H	[193]
		Prostate	High (serum, Primary)	H	[186,194]
		Lung	High (plasma)	H	[195,196]
		Melanoma	High (serum)	H	[197]
Growth differentiation factor 15/GDF-15	Important role in regulation of inflammatory pathways by inhibition of macrophages and involvement in regulation of apoptosis, cell growth and cell repair	Colorectal	High (serum, Primary)	CL, H	[198,199]
		Breast	Suppression promotes metastasis	CL, H	[200]
		Pancreas	High (plasma, Primary)	CL, H	[201]
		Liver	High (Primary)	H	[202]
		Prostate	High (serum)	CL, H	[199]
		Lung	Correlates with metastasis	CL, M	[203]
		Melanoma	High (serum)	H	[204]
Vascular endothelial growth factor/VEGF	Involved in angiogenesis and normal physiological function, development, wound healing, hematopoiesis	Colorectal	High mRNA (Primary) High (serum)	H	[205,206]
		Breast	High (Primary), High (serum)	H	[207,208]
		Pancreas	High (Primary) Correlation with TNM	H	[209]
		Liver	High (serum)	H	[210]
		Prostate	High (plasma, Primary)	H	[211]
		Lung	High (serum)	H	[212]
		Melanoma	High (serum, CL)	CL, H	[213]
Neuregulin 4/NRG-4	Potential activity in inhibition of lipogenesis in liver and control of glucose and lipid homeostasis	Colorectal	Induces cell survival in cells	CL, M	[214]
		Breast	High (Primary)	H	[215]
		Pancreas	Normal pancreas function	H	[216]
		Liver	Protection against IR	M, H	[217]
		Prostate	High (advanced stages)	H	[218]
		Lung	N/A		
		Melanoma	High (Primary, CL)	H	[219]

^1^ Abbreviations: Primary: Primary Tumors; H: Human; CL: Cell Line; M: Mice; N/A: Not available data.

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
