# Peer review of "Emerging Roles for Browning of White Adipose Tissue in Prostate Cancer Malignant Behaviour"

_ijms, 2021, doi:10.3390/ijms22115560_

Round 1

Reviewer 1 Report

The manuscript by Álvarez-Artime is review paper about the role/roles of factors released by white and brown adipose tissue and how the factors affect prostate cancer initiation, progression and metastasis. There have been very few reviews on this topic, making any new contribution to the field interesting. Overall, the manuscript is well written, the tables are very helpful, and the literature review offers a useful overview of current research and policy, and the resulting bibliography provides a very useful resource for current practitioners.

I only have minor comments:

As constructive criticism, I have to say that a couple of schematic figures or illustrations would have helped the reader to better follow this review.

Line 74: phosphatidylinositol instead of phosphatidylinisotol

Table 1: Column “Beige”, please correct “Varibale” and “Miltiple”

Line 77: Please correct the title of chapter 3

Line 90: Please rephrase the sentence starting with “In addition, pharmacological activator also been described…..”

Line 117: Please delete the “point” at the end of the title

Line 124: It has been instead of it is been

Line 164-167: This sentence is hard to follow. Please rephrase it

Line 169: “a” scenario

Line 173: mesenchymal instead of mesenchymal

Line 177. A point is necessary at the end of the sentence

Line 211: On the contrary, in the prostate…..

Line 212: compared

Lines 213-214: This sentence is incorrect “Similar results have been reported in colon cancer, where an association between tumour development and progression”. Please rewrite it

Line 223: Collectively instead of Collectivley

Please, for coherence, you should use tumor or tumour

Line 309: see Line 117

Line 351: in reducing

Author Response

Attached please find the answers to the reviewer’s comments. We really appreciate the reviewer’s comments as they truly improve the quality of the new manuscript version. Therefore, we have tried to address every single detailed mentioned by reviewers in an attempt to clarify all the points raised by reviewers. We hope that the all the changes made will match reviewer’s critics. The English has been thoroughly checked and changed, as we have used the professional version of the ‘Grammarly’; also it has been checked by an expertise in English.

Answers to R1:

R1: As constructive criticism, I have to say that a couple of schematic figures or illustrations would have helped the reader to better follow this review.

Answer: We appreciate this critical comment and accordingly we have improved the figure trying to include all the molecular aspects mentioned in the review. We hope that the reviewer will find it adequate in the new version

 As for the rest of comments, we have changed all the minor details mentioned by reviewer in lines 74, 77, 90, 117, 124, 164-7, 169, 173, 177, 211, 212, 213-14, 223, 309 and 351, as well as the table 1. Again, thank you for the thoroughly revision

Reviewer 2 Report

The manuscript by Álvarez-Artime et al described a review of the literature on the role of different factors released by white and brown adipose tissue which could participate on the initiation, progression and metastasis and resistance to chemotherapy of cancer and particularly of prostate cancer.

The manuscript is interesting and well-designed. However, I suggest some modifications to ameliorate the readability and the impact of this review.

1) make the title more focused: I suggest “Emerging Roles for Browning of White Adipose Tissue in Prostate cancer malignant behaviour”

2) enrich the manuscript (page 4 lane 174) with some recently published studies on the crosstalk between adipose tissue and prostate cancer

3) strengthen the evidence on the role of adipose tissue in other endocrine-related cancer such as breast (lane 228 page 5).

4) Add a figure representing the most important mechanisms proposed as drivers of the effect of adipose tissue on growth, migration and invasion and chemoresistance in cancer cells.

5) Many grammatical errors are present throughout the text:

page 1 Lane 41 not gen but gene

Page 4 lane 140 not asocciated but associated

Page 4 lane 145 not adittion but addition

Page 4 lane 165 not backfeed but feedback

Page 4 lane 169 not an scenario but a scenario

Page 5 lane 209 not a risk prognostic factor but a prognostic risk factor

Page 6 lane 267 not obsessed but obese

Page 10 lane 375 not fuelling but fueling.

Please, ask for English editing by a native speaker.

Author Response

Answer to R2:

R2: make the title more focused: I suggest “Emerging Roles for Browning of White Adipose Tissue in Prostate cancer malignant behaviour”.

Answer: We appreciate the criticism and we have changed the title accordingly.

R2: enrich the manuscript (page 4 lane 174) with some recently published studies on the crosstalk between adipose tissue and prostate cancer…

Strengthen the evidence on the role of adipose tissue in other endocrine-related cancer such as breast (lane 228 page 5).

Answer: We have included new information and references as requested by reviewer in both cases, including also new information about breast cancer and adipose tissue

R2:  Add a figure representing the most important mechanisms proposed as drivers of the effect of adipose tissue on growth, migration and invasion and chemoresistance in cancer cells.

As we mentioned to R1, who also highlighted this issue, we have included more information in Figure 1, trying to cover all the molecular aspects mentioned in the review.

R2: Many grammatical errors are present throughout the text:

page 1 Lane 41 not gen but gene

Page 4 lane 140 not asocciated but associated

Page 4 lane 145 not adittion but addition

Page 4 lane 165 not backfeed but feedback

Page 4 lane 169 not an scenario but a scenario

Page 5 lane 209 not a risk prognostic factor but a prognostic risk factor

Page 6 lane 267 not obsessed but obese

Page 10 lane 375 not fuelling but fueling.

Please, ask for English editing by a native speaker.

Answer: We have changed all the typos indicated and we have made a thoroughly revision of the English. In addition to the professional Grammarly version, the manuscript has been revised by an expert and several grammar and vocabulary changes have been made.
